# The Effect of Perioperative Administration of Probiotics on Colorectal Cancer Surgery Outcomes

**DOI:** 10.3390/nu13051451

**Published:** 2021-04-25

**Authors:** Louise Pitsillides, Gianluca Pellino, Paris Tekkis, Christos Kontovounisios

**Affiliations:** 1Department of Surgery and Cancer, South Kensington Campus, Imperial College London, London SW7 2AZ, UK; louise.pitsillides16@imperial.ac.uk (L.P.); p.tekkis@imperial.ac.uk (P.T.); 2Department of Advanced Medical and Surgical Sciences, Università degli Studi della Campania Luigi Vanvitelli, 80138 Naples, Italy; gipe1984@gmail.com; 3Colorectal Surgery, Vall d’Hebron University Hospital, 08035 Barcelona, Spain; 4Department of Colorectal Surgery, Chelsea and Westminster Hospital, London SW10 9NH, UK; 5Department of Colorectal Surgery, Royal Marsden Hospital, London SW3 6JJ, UK

**Keywords:** probiotics, colorectal cancer, perioperative care, colonic resection, colorectal surgery

## Abstract

The perioperative care of colorectal cancer (CRC) patients includes antibiotics. Although antibiotics do provide a certain protection against infections, they do not eliminate them completely, and they do carry risks of microbial resistance and disruption of the microbiome. Probiotics can maintain the microbiome’s balance postoperatively by maintaining intestinal mucosal integrity and reducing bacterial translocation (BT). This review aims to assess the role of probiotics in the perioperative management of CRC patients. The outcomes were categorised into: postoperative infectious and non-infectious complications, BT rate analysis, and intestinal permeability assessment. Fifteen randomised controlled trials (RCTs) were included. There was a trend towards lower rates of postoperative infectious and non-infectious complications with probiotics versus placebo. Probiotics reduced BT, maintained intestinal mucosal permeability, and provided a better balance of beneficial to pathogenic microorganisms. Heterogeneity among RCTs was high. Factors that influence the effect of probiotics include the species used, using a combination vs. single species, the duration of administration, and the location of the bowel resection. Although this review provided evidence for how probiotics possibly operate and reported notable evidence that probiotics can lower rates of infections, heterogeneity was observed. In order to corroborate the findings, future RCTs should keep the aforementioned factors constant.

## 1. Introduction

Colorectal cancer (CRC) is the third most common cancer in the world [1]. Its incidence has been increasing globally in the past few years [2]. Management of CRC consists mainly of surgical resection of the tumour with neoadjuvant or adjuvant chemotherapy [3]. Even though a number of preoperative measures are taken to minimise the risks of postoperative infections and complications, they do not eliminate them completely. Prophylactic antibiotics do confer some protection against infections but cause dysbiosis, and this, along with the trauma and stress that the surgery imposes, allow pathogenic bacteria to undergo bacterial translocation (BT) to distant locations and organs, hence causing infections [4,5,6].

There is evidence that probiotics can maintain the intestinal membrane’s integrity and, in this way, can reduce the rates of BT [7]. Probiotics can also enhance the balance of beneficial to pathogenic bacteria by competitively adhering to mucosal membranes [8]. By doing so, probiotics can effectively decrease the rate of infections after colorectal surgery, namely surgical site infections (SSIs), abscesses, and anastomotic leaks.

Other abdominal procedures such as pancreatic duodenectomies, biliary cancer surgery, upper gastrointestinal surgery, and liver transplants have reported positive results of perioperative probiotic use [9,10,11,12]. Probiotics have also been successfully used for the management of chemotherapy-induced diarrhoea and in some patients with inflammatory bowel diseases (IBD) [13,14,15].

The aim of this review is to evaluate how the use of probiotics perioperatively in CRC surgery can affect surgical outcomes.

## 2. Materials and Methods

This is a Preferred Reporting Items for Systematic Reviews and Meta-Analyses (PRISMA) compliant review. A literature search was conducted by (LP) with the Imperial College Library services using the Embase, Medline, and Cochrane databases for articles published between January 2000 and January 2020. 

### 2.1. Search Strategy

Search terms used included: Probiotics, Colorectal cancer, *Lactobacillus*, and *Bifidobacterium*. The terms were used in all possible combinations to retrieve the maximal number of articles. Keywords and MeSH terms used in the database search can be found in Table 1. Additional articles identified by a manual search of reference lists from the extracted papers and other reviews were also included.

### 2.2. Study Selection and Quality Assesment

Eligibility criteria for this review were RCTs that involved humans, colorectal cancer procedures, the usage of probiotics perioperatively, and comparisons between a group given probiotics and a control group. Studies using synbiotics or prebiotics, studies on non-colorectal cancer procedures, and animal studies were excluded. English language restrictions were imposed. Outcomes of interest included: SSIs, anastomotic leaks, incidence of fever, incidence of ileus, day to first flatus and stool, duration of abdominal distention and cramps postoperatively, and the effect of probiotics on the bacterial translocation and intestinal permeability in the probiotics group compared to control groups.

An initial assessment by title and abstract using the computer software EndNoteX9 (Clarivate; USA, UK) highlighted studies of potential interest. The full-text copies of each of these papers were reviewed by (LP). For any discrepancies and disagreements regarding paper inclusions, there were discussions between two authors (LP and CK). 

In order to assess the risk of bias in the RCTs selected, a quality assessment was carried out using the revised Cochrane Risk of Bias tool. The tool assesses the risk of bias based on different bias domains, including trial design, conduct, and reporting. None of the RCTs included had a high risk of bias, so all were included [16].

### 2.3. Data Extraction and Analysis

Data of the final selected papers was extracted by LP to include author, year of publication, number of patients in each group, outcomes measured, probiotic species in regimen, and duration of probiotics regimen.

To summarise postoperative outcomes and complications data, descriptive statistics such as means were included in the results tables. Due to heterogeneity in describing effects on BT and intestinal permeability, a selective descriptive analysis was included instead.

## 3. Results

The initial search yielded 2718 search results. A manual bibliographic search obtained another six studies that were not extracted through the online database search. A total of 1909 studies were either automatically or manually removed as they were duplicates. Using EndNoteX9 (Alfasoft Limited, Luton Bedfordshire, UK), the 1909 studies were screened, and 1884 records were excluded due to being irrelevant to the topic, in a foreign language, or in a non-full-text format. The remaining 25 articles were closely assessed, and a further 10 were removed. A total of 15 articles were included. The flow diagram of the research is presented in Figure 1.

### 3.1. Postoperative Infectious Complications

Colorectal resections have a significant rate of postoperative complications. Most SSIs are superficial, deep incisional, or occur in the organ/space (CDC) [17]. The outcomes of the infectious postoperative complications of the RCTs included in this review can be found in Table 2.

### 3.2. Postoperative Non-Infectious Complications and Outcomes

Colorectal specific postoperative non-infectious complications and outcomes are summarised in Table 3.

### 3.3. Bacterial Translocation and Intestinal Permeability 

A key mechanism of many postoperative complications in colorectal surgeries is an increased intestinal permeability, which in turn increases the rate of BT to distant sites. Many of the RCTs included in this review assessed perioperative BT and intestinal permeability rates. 

Liu et al. [22,29] assessed the amount of BT to the mesenteric lymph nodes (MLNs). In 2011 [22], the probiotic group had a significantly lower postoperative incidence of BT compared to the placebo (18% and 28%, respectively (*p* < 0.01)). In 2013 [29], there was a significantly lower postoperative incidence of BT in the group of probiotics compared to the placebo (13% and 28%, respectively (*p* = 0.027)).

Mangell et al. [24] also assessed the amount of BT to MLNs. They found that the probiotics group had higher rates of BT than the placebo, but this was not significant (27% and 13%, respectively (*p* = 0.374)).

Liu et al. [22,29] assessed intestinal permeability using the urine lactulose/mannitol (L/M) test. It was performed preoperatively once and on days +3 and +10. In 2011 [20], on day +3, both groups had a significant increase from before surgery (0.23 ± 0.08 to 0.17 ± 0.04 and 0.23 ± 0.06 to 0.19 ± 0.05, respectively (*p* < 0.05)). On day/+10, the L/M ratio was significantly lower in the probiotic group than the placebo group (0.18 ± 0.03 and 0.22 ± 0.04, respectively (*p* = 0.04)). In 2013 [29], the L/M ratio was similar in both groups on day/+3 (0.222 ± 0.059 and 0.231 ± 0.058, respectively (*p* = 0.348)). On day +10, the mean L/M ratio in the probiotics group was significantly lower than in the placebo group (0.166 ± 0.039 and 0.216 ± 0.061, respectively (*p* = 0.001)). This study also compared all of their sample results to serum zonulin levels of the same day samples. There was a direct correlation between postoperative zonulin levels and L/M ratio (r = 0.504, *p* = 0.001).

Liu et al. [22,29] assessed the mucosal to serosal transport of macromolecules by measuring the trans-epithelial flux of horseradish peroxidase (HRP). In their 2011 study [20], they reported that the mean colon mucosal trans-epithelial resistance (TER) was significantly higher in the probiotics group (18.4 ± 5.1 Ω/cm^2^ and 13.7 ± 4.2 Ω/cm^2^, respectively (*p* < 0.05)).

In their 2013 [27] study, they reported the mean colon mucosal TER in the probiotic group to be significantly higher than that in the placebo group (19.21 ± 6.02 Ω/cm^2^ and 12.66 ± 5.86 Ω/cm^2^, respectively (*p* = 0.001)). When comparing these results to postoperative zonulin levels, a direct correlation was found between the serum zonulin concentration and the TER (r = 0.900, *p* = 0.037) and also between the serum zonulin concentration and the HRP flux (r = 1.000, *p* = 0.001).

Liu et al. [22] performed immunohistochemistry assays and fluorescence staining under confocal laser scanning microscopes to determine the expression of claudin-1, JAM-1, and occludin, which are all tight junction (TJ) proteins. The placebo group had a substantial loss of claudin-1, JAM-1, and occludin from the TJs.

Liu et al. [29] measured the serum zonulin concentration 24 h postoperatively. The placebo group had a significantly higher zonulin concentration than placebo (1.08 ± 0.28 ng/mg and 0.39 ± 0.26 ng/mg of protein, respectively (*p* = 0.001)).

Zhang et al. [23] assessed the serum endotoxin, D-lactic acid levels, and *E. Coli* detection rate. There was a significant increase in the D-lactic acid and endotoxin levels before and after surgery in the placebo group (*p* < 0.001), but there was no significant change in the probiotics group. The probiotic group had overall lower levels of endotoxin and D-lactic acid than the placebo group (*p* < 0.001). The detection rate of *E. Coli* was significantly higher in the placebo group than the probiotic group (26.7% and 3.3%, respectively (*p* = 0.026)).

## 4. Discussion

It is important to seek strategies to reduce postoperative complications as they increase the length of stay and medical costs and cause great discomfort to patients [25,32,33]. Perioperative probiotic administration could possibly minimise the risk for postoperative complications.

There is evidence that the intestinal microbiota has a direct effect on SSIs rates, and in colorectal surgeries, there is an increase in dysbiosis due to surgical trauma and antibiotics administration, which both disrupt the microbiome [34]. Intra-abdominal abscesses and SSIs were lower in the probiotics group in most of the trials included in this review.

Ileus is relatively common after colorectal surgery and may cause several issues. Consequently, probiotics can possibly decrease postoperative ileus by controlling bacterial metabolites and pH levels, as suggested by the results of the trial in this review [35,36,37].

Overall, the groups receiving probiotics in all the RCTs had a shorter length of stay in hospital. The shorter the stay in hospital, the less the impact it has on medical costs. Additionally, the less time a patient spends in hospital, the smaller their risk is of contracting nosocomial infections [21].

CRC resections are highly traumatic surgeries and can lead to mucosal barrier dysfunction [38]. This can damage the intestinal mucosal barrier and increase BT to extra-intestinal sites. This ultimately means that pathogenic bacteria and endotoxins can cause distant infections [39]. A few studies in this review investigated BT rates and intestinal permeability and found that the group receiving probiotics had a lower incidence of BT to distant sites such as MLNs. The same studies also reported lower rates in some postoperative outcomes, including: SSIs, diarrhoea, fever, bacteraemia and septicaemia, abdominal cramps and distention, and a lower mean length of hospital stay in the probiotics group.

A possible explanation as to why probiotics may help to reduce BT is by maintaining the intestinal mucosa’s integrity and preventing barrier dysfunction [40]. In their 2011 study, Liu et al. [22] assessed the TJ barrier function by measuring changes in the distribution of specific TJ proteins and/or their levels of expression. They demonstrated that probiotics could stabilise cellular claudin-1, JAM-1, and occludin, which are prone to damage during surgical manipulations. The same study also demonstrated a lower BT incidence, lower blood bacterial positive rate, and lower microbial DNA positive rate in the probiotics group. This means that probiotics have the ability to alleviate intestinal villous injury, which eventually leads to lower distant infectious rates [22].

In surgical studies up to now, the most commonly used probiotic species are Lactobacilli and Bifidobacteria [26]. In this review, the three most commonly used species were: *Lactobacillus acidophilus*, *Bifidobacterium longum*, and *Lactobacillus plantarum* (Figure 2). The presence of beneficial bacteria in the gut, such as *Bifidobacteria* and *Lactobacilii*, is enhanced when administrating the correct probiotics. Probiotics play an important role in stabilising this microbiological environment [41].

The average number of species used in the regimen of the RCTs in this review is 4 (Table 4). Using multiple species in a probiotic regimen is reported to improve treatment and surgical recovery [42]. Yang et al. [27], however, argued that there is no evidence that multiple strains used together can have more favourable effects than a single strain. It is important to note that this field has a lack of dose–response studies, which are essential to establish proper recommendations in order to influence clinical guidelines [21].

Evidence from studies using probiotics in other abdominal surgeries, such as pancreatic duodenectomies, biliary cancer surgeries, upper gastrointestinal surgeries, and liver transplants, supports the notion that probiotics can maintain the microbiota balance [11,12,13,14,26]. Probiotics have been successfully used for the treatment of IBD and in the chemoprophylaxis of gastrointestinal cancer [13,14].

Even though there is sufficient evidence that probiotics may help to reduce postoperative complications, this is not always the case. Some studies in this review showed no significant reduction in the incidence of complications postoperatively in the probiotics group. Other studies involving general surgery patients also showed no difference in the use of probiotics [24,43]. Some studies have even reported that probiotics can increase the rate of BT, hence increasing the risk of complications postoperatively [43].

A factor that could impact how probiotics affect postoperative outcomes could be the location of the colonic resection. A possible way around this is to administer different types of probiotics depending on where the colonic resection will be. Moving down the colon, the environment changes; it becomes more acidic and higher in oxygen, meaning that different types of commensal bacteria survive better in each part [44,45]. Administrating probiotics with the most prevalent commensal bacteria according to where the resection will be ensures that those species are the most efficient for that location. Furthermore, direct jejunal administration of probiotics could skip gastric acids and pancreatic enzymes and ensure the safe arrival of the microorganisms in the colon.

There are significant discrepancies in the exact duration of administration of probiotics in the RCTs in this review. This can provide a possible explanation as to why some trials did not report many significant differences between the two groups. However, a colonoscopy study showed no difference in the bacterial composition of the colon when Lactobacilli were administered for different durations [45].

There is significant heterogeneity between the RCTs included in this review. Using specific inclusion criteria, manually extracting and screening the methodology of each paper before including them, and a quality assessment using the Revised Cochrane Risk of Bias tool were the measures taken to minimise the heterogeneity. Even after implementing these measures, the heterogeneity is still significant, which made it difficult to group and compare the results. This highlights the need for consistency between future RCTs on this topic in order to make generalisable conclusions easier and to a higher standard.

This review recommends that a RCT with a large sample size should take place to confirm the positive benefits of probiotics. Postoperative complications of both natures should be recorded along with analysing MLNs to assess BT as it is the most direct and representative measurement of BT [46]. A length of 7 days preoperatively and 7 days postoperatively should be considered as the duration of administration. An ideal combination of probiotics to be used is mainly Bifidobacteria and Lactobacilli. Location of resections should be noted, and the probiotic regimen could be altered accordingly.

## 5. Conclusions

Probiotics have shown promising results in terms of reducing BT, enhancing intestinal mucosal integrity, hindering barrier dysfunction, and preserving a favourable balance of beneficial to pathogenic bacteria in the gut. These factors can ultimately reduce the risk for distant infections and other non-infectious complications after colorectal surgery.

Due to the heterogeneity between the RCTs included in this review, no definite conclusions can be drawn. However, most infectious and non-infectious complications did have a lower incidence rate in the probiotics group, even though this was not statistically significant in most cases.

A larger RCT that accounts for the duration of administration, combination of probiotics species in the regime, and location of colorectal resection should be used to deduce whether probiotics do efficiently reduce postoperative complications and whether they should be integrated into clinical practice.

## Figures and Tables

**Figure 1 nutrients-13-01451-f001:**
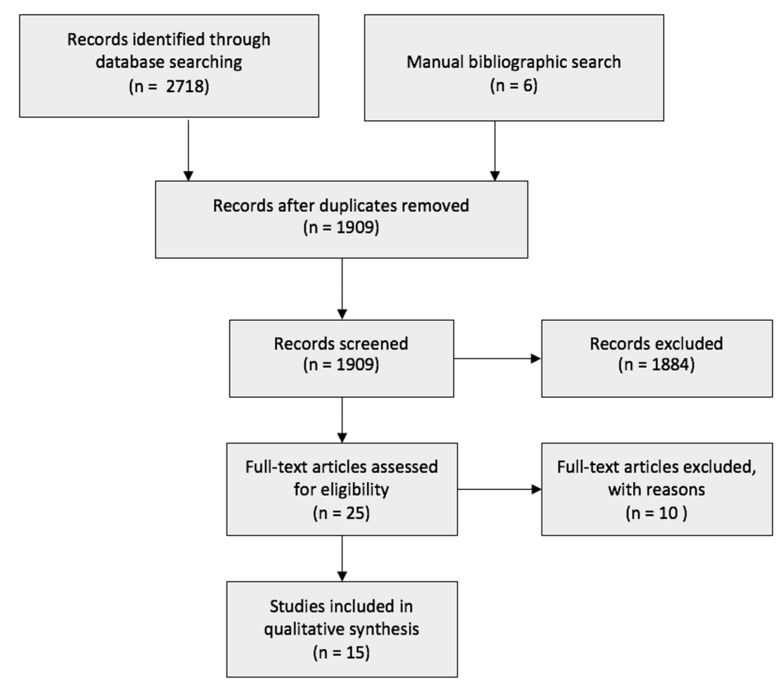
Flowchart of the search strategy.

**Figure 2 nutrients-13-01451-f002:**
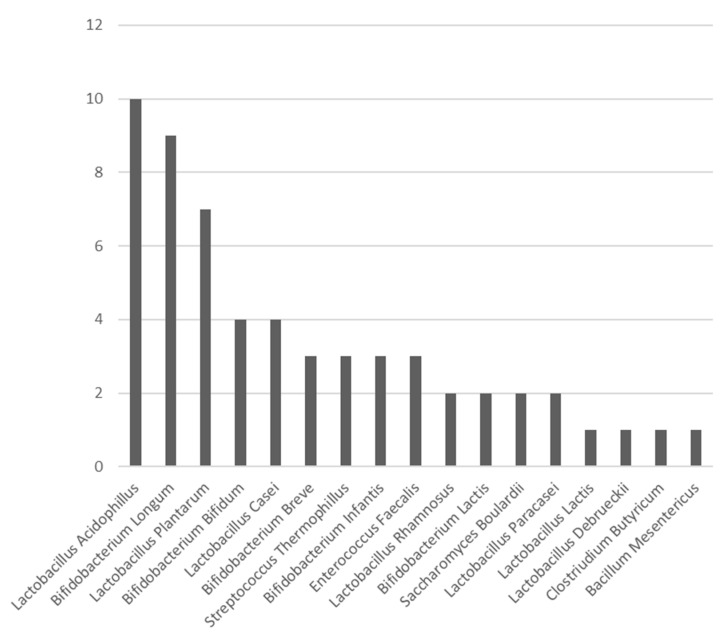
Bar chart exhibiting the number of times each probiotic species was used in the RCTs included in this study.

**Table 1 nutrients-13-01451-t001:** Keywords and MeSH terms used to search Embase, Medline, and Cochrane through the dates 2–10 February 2020.

Key Words	MESH Terms
Probiotic(s), *Lactobacillus*/*i*, *Bifidobacteria*/*um* Colorectal cancer/tumour/neoplasm/carcinoma, Rectal cancer/tumour/neoplasm/carcinoma, Colon cancer/tumour/neoplasm/carcinoma	Probiotics, *Lactobacillus*, *Bifidobacterium*, Colorectal neoplasms, Colonic neoplasms, Rectal neoplasm

**Table 2 nutrients-13-01451-t002:** Postoperative infectious complications incidence.

**Author (Year)**	**Number of Participants (Probiotics + Placebo)**	**SSIs Rate in Probiotics Group (%)**	**SSIs Rate in Placebo Group (%)**	***p*-Value**
Aisu (2015) [18]	156 (75 + 81)	6.7	19.7	0.016 *
Kotzampassi (2015) [19]	164 (84 + 80)	6.0	16.0	0.040 *
Bajramagic (2019) [8]	78 (39 + 39)	28.2	35.9	0.682
Kakaei (2019) [20]	99 (50 + 49)	6.0	10.0	0.460
Tan (2016) [21]	40 (20 + 20)	5.0	10.0	0.548
Liu (2011) [22]	100 (50 + 50)	7.0	11.0	>0.05
Zhang (2012) [23]	60 (30 + 30)	3.3	13.3	0.353
Mangell (2012) [24]	72 (36 + 36)	3.0	3.0	N/A
Sadahiro (2014) [25]	310 (100 + 95) (third group taking antibiotics: 99)	18.9	17.9	1.00
**Author (Year)**	**Number of Participants (Probiotics + Placebo)**	**Organ/Space SSIs Rate in Probiotics Group (%)**	**Organ/Space SSIs Rate in Placebo Group (%)**	***p*-Value**
Aisu (2015) [18]	156 (75 + 81)	2.7	4.9	0.016 *
Sadahiro (2014) [25]	310 (100 + 95) (third group taking antibiotics: 99)	4	5.3	0.93
**Author (Year)**	**Number of Participants (Probiotics + Placebo)**	**Intra-Abdominal Abscess Rate in Probiotics Group (%)**	**Intra-Abdominal Abscess Rate in Placebo Group (%)**	***p*-Value**
Bajramagic (2019) [8]	78 (39 + 39)	12.8	17.9	0.530
Consoli (2016) [26]	33 (15 + 18)	0	4.0	>0.10
Mangell (2012) [24]	72 (36 + 36)	0	3.0	N/A
Zhang (2012) [23]	60 (30 + 30)	6.7	3.3	1.00
**Author (Year)**	**Number of Participants (Probiotics + Placebo)**	**Anastomotic Leaks in Probiotics Group (%)**	**Anastomotic Leaks IN Placebo Group (%)**	***p*-Value**
Kotzampassi (2015) [19]	164 (84 + 80)	1.2	8.8	0.031 *
Sadahiro (2014) [25]	310 (100 + 95) (third group taking antibiotics, 99)	7.4	12.0	0.560
Yang (2016) [27]	60 (30 + 30)	3.3	6.7	1.00
Mizuta (2016) [28]	60 (31 + 29)	9.7	17.2	>0.05
Tan (2016) [21]	40 (20 + 20)	5.0	10.0	0.548
Zhang (2012) [23]	60 (30 + 30)	0	3.3	0.492
Mangell (2012) [24]	72 (36 + 36)	0	3.0	N/A
**Author (Year)**	**Number of Participants (Probiotics + Placebo)**	**Mean Incidence of Fever in Probiotics Group (days ± standard deviation)**	**Mean Incidence of Fever in Placebo Group (days ± standard deviation)**	***p*-Value**
Liu (2011) [22]	100 (50 + 50)	6.0 ±1.9	7.2 ± 2.1	<0.05 *
Liu (2013) [29]	138 (70 + 68)	5.82 ± 1.98	6.68 ± 2.29	0.015 *
Yang (2016) [27]	60 (30 + 30)	1.80 ± 2.34	4.77 ± 1.79	0.951

* = *p*-value is less than <0.05.

**Table 3 nutrients-13-01451-t003:** Incidence of postoperative non-infectious complications and outcomes.

**Author (Year)**	**Number of Participants (Probiotics + Placebo)**	**Incidence of Ileus in Probiotics Group (%)**	**Incidence of Ileus in Placebo Group (%)**		***p*-Value**
Bajramagic (2019) [8]	78 (39 + 39)	2.6	23.1		0.007 *
**Author (Year)**	**Number of Participants (Probiotics + Placebo)**	**Mean day to First Flatus in Probiotics Group (days ± standard deviation)**		**Mean Day to First Flatus In Placebo Group (days ± standard deviation)**	***p*-Value**
Aisu (2015) [18]	156 (75 + 81)	2.0 ± 1.1		2.8 ± 2	0.001 *
Yang (2016) [27]	60 (30 + 30)	3.27 ± 0.58		3.63 ± 0.67	0.0274 *
Mangell (2012) [24]	72 (36 + 36)	Median day to first flatus		Median day to first flatus	N/A
		2		3	
**Author (Year)**	**Number of Participants (Probiotics + Placebo)**	**Mean Day to First Stool in Probiotics Group (days ± standard deviation)**		**Mean Days to First Stool in Placebo Group (days ± standard deviation)**	***p*-Value**
Kotzampassi (2015) [19]	164 (84 + 80)	Lower ^a^		Higher ^a^	0.001 *
Yang (2016) [27]	60 (30 + 30)	3.87 ± 1.17		4.53 ± 1.11	0.0268 *
Mangell (2012) [24]	72 (36 + 36)	Median day to first stool		Median day to first stool	N/A
		4		4	
**Author (Year)**	**Number of Participants (Probiotics + Placebo)**	**Incidence of Abdominal Distention in Probiotics Group (%)**	**Incidence of Abdominal Distention in Placebo Group (%)**		***p*-Value**
Liu (2011) [22]	100 (50 + 50)	21.0	36.0		<0.05 *
Yang (2016) [27]	60 (30 + 30)	30.0	43.3		<0.05 *
**Author (Year)**	**Number of participants (probiotics + study)**	**Incidence of Abdominal Cramps in probiotics group (%)**	**Incidence of Abdominal Cramps in placebo group (%)**		***p*-Value**
Liu (2011) [22]	60 (30 + 30)	26.0	39.0		<0.05 *
**Author (Year)**	**Number of participants (probiotics + placebo)**	**Incidence of Diarrhoea in probiotics group (%)**	**Incidence of Diarrhoea in placebo group (%)**		***p*-Value**
Liu (2011) [22]	100 (50 + 50)	17.0	34.0		<0.05 *
Liu (2013) [29]	128 (70 + 68)	14.7	29.3		0.03 *
Yang (2016) [27]	60 (30 + 30)	26.7	53.3		0.0352 *
**Author (Year)**	**Number of Participants (Probiotics + Placebo)**	**Mean length of Stay in Probiotics Group (days ± standard deviation)**	**Mean Length of Stay in Placebo Group (days ± standard deviation)**		***p*-Value**
Kotzampassi (2015) [19]	164 (75 + 81)	Median length of stay	Median length of stay		<0.0001 *
		8	10		
Consoli (2016) [26]	33 (15 + 18)	10	11		>0.10
Kakaei (2019) [20]	99 (50 + 49)	5.96 ± 2.53	6.10 ± 2.44		0.30
Liu (2011) [22]	100 (50 + 50)	12.3 ± 2.3	12.6 ± 3.3		>0.05
Mizuta (2016) [28]	60 (31 + 29)	21.4 ± 10.1	23.0 ± 13.8		>0.05
Pellino (2013) [30]	18 (10 + 8)	12.0 ± 8.3	13.5 ± 4.8		>0.05
Stephens (2012) [31]	38 (20 + 18)	5.60 ± 2.93	6.45 ± 7.50		0.564
Zhang (2012) [23]	60 (30 + 30)	12.0 ± 3.0	14.0 ± 3.0		0.109
Yang (2016) [27]	60 (30 + 30)	15.86 ± 4.92	15.0 ± 4.31		0.487

^a^ = Kotzampassi et al. [19] provided no raw values for mean day to first stool, but the cumulative logarithmic graph illustrated a lower value for the probiotics group overall than the placebo, and this was statistically significant (*p* = 0.001). * = *p*-value is less than <0.05.

**Table 4 nutrients-13-01451-t004:** Probiotic composition and the number of probiotic strains used in each RCT included in this study.

Study	Probiotic Strains	No. of Strains	Duration of Administration
Bajramagic	*Lactobacillus casei*, *Lactobacillus acidophillus*, *Lactobacillus plantarum*, *Lactobacillus rhamnosus*, *Bifidobacterium lactis*, *Bifidobacterium bifidum*, *Bifidobacterium Breve*, *Streptococcus Thermophillus*	8	+3 to +30
Pellino	*Streptococcus thermophilus*, *Bifidobacterium longum*, *Bifidobacterium breve*, *Bifidobacterium infantis*, *Lactobacillus acidophillus*, *Lactobacillus plantarum*, *Lactobacillus paracasei*, *Lactobacillus debrueckii*	8	+0 to +28
Stephens	*Streptococcus thermophilus*, *Bifidobacterium longum*, *Bifidobacterium breve*, *Bifidobacterium infantis*, *Lactobacillus acidophillus*, *Lactobacillus plantarum*, *Lactobacillus paracasei*, *Lactobacillus debrueckii*	8	+0 to +28
Tan	*Lactobacillus acidophillus*, *Lactobacillus casei*, *Lactobacillus lactis*, *Bifidobacterium bifidum*, *Bifidobacterium longum*, *Bifidobacterium infantis*	6	−7 to +0
Kakaei	*Lactobacillus casei*, *Lactobacillus acidophillus*, *Bifidobacterium breve*, *Bifidobacterium longum*, *Streptococcus thermophilus*	5	−7 to +23
Yang	*Bifidobacterium longum*, *Lactobacillus acidophillus*, *Enterococcus faecalis*	5	−5 to +7
Kotzampassi	*Lactobacillus acidophillus*, *Lactobacillus plantarum*, *Bifidobacterium lactis*, *Sacharomyces boulardii*	4	−1 to +14
Aisu	*Enterococcus faecalis*, *Clostridium butyricum*, *Bacillum mesentericus*	3	−15/−3 ^a^
Liu 2011	*Lactobacillus plantarum*, *Lactobacillus acidophillus*, *Bifidobacterium longum*	3	−6 to +10
Liu 2013	*Lactobacillus plantarum*, *Lactobacillus acidophillus*, *Bifidobacterium longum*	3	−6 to +10
Zhang	*Bifidobacterium longum*, *Lactobacillus acidophillus*, *Enterococcus faecalis*	3	−5 to +3
Sadahiro	*Bifidobacterium bifidum*	1	−8 to −2 and +5 to +15
Consoli	*Sachharomyces boulardii*	1	−7 to +0 ^b^
Mangell	*Lactobacillus plantarum*	1	−8 to +0 and +1 to +6
Mizuta	*Bifidobacterium longum BB536*	1	−14/−7 to +14

^a^ = restarted when the patient started drinking water (duration not stated), ^b^ = post-operative duration not stated.

## Data Availability

All data reported in this study can be found in the original articles as referenced.

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
