# Peer review of "The Effect of Perioperative Administration of Probiotics on Colorectal Cancer Surgery Outcomes"

_nutrients, 2021, doi:10.3390/nu13051451_

Round 1
Reviewer 1 Report
This is a generally well written review analyzing proposed beneficial effects of perioperative probiotics in colorectal cancer surgery. Unfortunately, there is a considerable heterogeneity among the included Rct:s making peopled analyses and generalizable conclusions difficult. Search algorithms and processing of results adhere to modern standards. This fact does however not decrease the need for good review articles but emphasizes the need for highlighting and briefly discussing the quality of the included studies and to point out possible reasons for the disparity in results. I would recommend the authors to extend the discussion by this dimension to further improve the manuscript.
Author Response
Response to Reviewer 1 Comments
Point 1:
This is a generally well written review analyzing proposed beneficial effects of perioperative probiotics in colorectal cancer surgery. Unfortunately, there is a considerable heterogeneity among the included Rct:s making peopled analyses and generalizable conclusions difficult. Search algorithms and processing of results adhere to modern standards. This fact does however not decrease the need for good review articles but emphasizes the need for highlighting and briefly discussing the quality of the included studies and to point out possible reasons for the disparity in results. I would recommend the authors to extend the discussion by this dimension to further improve the manuscript.
Response 1:
Dear Editor, thank you for reviewing this manuscript and making the above suggestions. The authors of this paper have added this paragraph in the discussion as suggested in order to address this issue:
“There is significant heterogeneity between the RCTs included in this review. Using specific inclusion criteria, manually extracting and screening methodology of each paper before including them and a quality assessment using the Revised Cochrane Risk of Bias tool, where measures taken to minimise the heterogeneity. Even after implicating these measures the heterogeneity is still significant which made it hard to group and compare the results. This highlights the need for consistency between future RCTs on this topic in order to make generalizable conclusions easier and to a higher standard”
Found on page 13, lines 316-322.
Reviewer 2 Report
This is an interesting study. Nevertheless, I would like to address a number of suggestions to you and your colleagues.
Introduction
Page 1 line 32. Please add dot after reference one.
Page 1 line 33. It is generally accepted that the incidents have been increasing globally, but the reference that you used describes cases only from England. Please add the data about global statistics.
The Results section of the manuscript are incomplete and not well-balanced.
As the authors indicated "The aim of this review is to evaluate how the use of probiotics peri-operatively in CRC 51 surgery, can affect surgical outcomes.", unfortunately, this article does not analyze the role of probiotics in CRC.
Results sections contains three sub-sections:
3.1 Post-operative infectious complications
3.2 Post-operative non-infectious complications and outcomes
3.3. Bacterial Translocation and intestinal permeability
The description of role of probiotics in CRC is incorrectly prepared and contain only one table.
Page 7 line 167 and 171, please write the names of species in italics.
Page 9, line 151; Horseradish peroxidase (HRP) but not (HPO). Please correct.
Page 10 Figure 2: The quality of figure 2 is too low.
Page 10, table 4: Please write the names of species in italics.
Reference’s inconsistency
Your references number in the text are mismatched with number at the end of your article in references section. For example:
Liu et al 2011 [20] table 2, page 6, page 11, line 213, but your article's reference 20 is "Kakaei F, Shahrasbi M, et al. Assessment of probiotic effects on colorectal surgery complications: A double blinded, randomized clinical trial. Biomedical Research and Therapy."
2019;6(3):3067-3072
Tan (2016) [19] table 2, page 6, but it is reference number 21 "Tan CK, Said S, et al. Pre-surgical administration of microbial cell preparation in colorectal cancer patients: A randomized controlled trial. World Journals of Surgery. 2016;40(8):1985-1992"
Author Response
Response to Reviewer 2 Comments
Point 1: Page 1 line 32. Please add dot after reference one.
Response 1: Revised as requested. Page 1 line 32 dot added after reference one.
Point 2: Page 1 line 33. It is generally accepted that the incidents have been increasing globally, but the reference that you used describes cases only from England. Please add the data about global statistics.
Response 2: Thank you for suggesting this edit. A new reference has been added which describes data of global statistics on the increasing incidence of colorectal cancer. Reference number 2 on page 14: “Arnold M, Sierra MS, Laversanne M, Soerjomataram I, Jemal A, Bray F (2017) Global patterns and trends in colorectal cancer incidence and mortality. Gut 66:683–691”
Point 3: The Results section of the manuscript are incomplete and not well-balanced.
As the authors indicated "The aim of this review is to evaluate how the use of probiotics peri-operatively in CRC 51 surgery, can affect surgical outcomes.", unfortunately, this article does not analyze the role of probiotics in CRC.
Response 3: Dear Reviewer, thank you for this comment. To clarify the aim of this review was to evaluate how the perioperative administration of probiotics can affect any surgical outcomes and complications, post colorectal cancer resection surgeries. The aim of this paper was not to analyse the role of probiotics in Colorectal cancer. Therefore, no changes have been made according to point number 3 made by Reviewer 2.
Point 4: Results sections contains three sub-sections:
3.1 Post-operative infectious complications
3.2 Post-operative non-infectious complications and outcomes
3.3. Bacterial Translocation and intestinal permeability
The description of role of probiotics in CRC is incorrectly prepared and contain only one table.
Response 4: Dear Reviewer, thank you for this comment. The 2 tables in the result section, as labelled, summarise the post-operative outcomes of colorectal cancer resection surgeries. For section 3.1 “Post-operative infectious complication incidences”, data from the Randomised Controlled Trials (RCTs) included in this review, was extracted from each paper and grouped under common complications such as: Surgical Site Infections (SSIs), Organ/Space SSIs, Intra-Abdominal Abscesses, Anastomotic Leaks, Fever and Ileus. For section 3.2 “Post-operative non-infectious complications and outcomes”, data from the RCTs included in this review was extracted and grouped under common complications/outcomes as follows: Ileus, Mean day to first flatus, Mean day to first stool, Abdominal distention rates, Abdominal cramps rates, Diarrhoea and Mean length of stay in hospital. Finally, in the last section of the results, Section 3.3 “Bacterial Translocation and Intestinal Permeability”, these are two main mechanisms of distant site infections post-operatively in colorectal cancer surgeries by which probiotics have been described in the literature to effect to favourable outcomes. Data was extracted from the RCTs included in this review and rather than being summarised in tables, due to their heterogeneity they couldn’t be grouped so instead were described. To clarify this paper’s aim was not to explore the role of probiotics in colorectal cancer but rather, as described above, to evaluate how the perioperative administration of probiotics can affect post-operative outcomes and complications of colorectal cancer surgeries. Therefore, no changes have been made regarding Point 4 made by Reviewer 2.
Point 5: Page 7 line 167 and 171, please write the names of species in italics.
Response 5: Revised as requested. Where “E. Coli” is mentioned the format has been changed from normal to italics. Page 7 Lines 168 and 172
Point 6: Page 9, line 151; Horseradish peroxidase (HRP) but not (HPO). Please correct.
Response 6: Thank you for this suggestion it has been revised as requested: HPO changed to HRP on lines: 199, 207 and line 365 (abbreviations table).
Point 7: Page 10 Figure 2: The quality of figure 2 is too low.
Response 7: Revised as requested, A new image has been inserted with a higher pixel quality. Page 10 Figure 2.
Point 8: Page 10, table 4: Please write the names of species in italics.
Response 8: Revised as requested. Page 10 Table 4. The names of probiotics species have been changed to Italic format. (Column 2 “Probiotic Strains”).
Point 9: Your references number in the text are mismatched with number at the end of your article in references section. For example:
Liu et al 2011 [20] table 2, page 6, page 11, line 213, but your article's reference 20 is "Kakaei F, Shahrasbi M, et al. Assessment of probiotic effects on colorectal surgery complications: A double blinded, randomized clinical trial. Biomedical Research and Therapy."
Tan (2016) [19] table 2, page 6, but it is reference number 21 "Tan CK, Said S, et al. Pre-surgical administration of microbial cell preparation in colorectal cancer patients: A randomized controlled trial. World Journals of Surgery. 2016;40(8):1985-1992"
Response 9: Revised as requested. Furthermore, the author LP of this manuscript apologises for this discrepancy. The table references have all been corrected. Please see tracked changes in all tables in the results section.
Round 2
Reviewer 1 Report
The authors have adequately addressed the issue raised in my review report.
Reviewer 2 Report
All my previous remarks for this article were corrected by authors